# Reduced surface fine dust under droughts over the southeastern United States during summertime: observations and CMIP6 model simulations

Wei Li[1] and Yuxuan Wang[1]

[1]Department of Earth and Atmospheric Sciences, University of Houston, Houston, Texas, USA.

*Corresponding author:* Yuxuan Wang (ywang246@central.uh.edu)

**Abstract.** Drought is an extreme hydroclimate event that has been shown to cause the increase of surface fine dust near source regions, yet the drought-dust relationship in regions predominantly influenced by long-range transported dust such as the southeastern US (SEUS) has received less attention. Using long-term surface fine dust observations, weekly US Drought Monitor (USDM), and monthly Standardized Precipitation-Evapotranspiration Index (SPEI), this study unmasks spatial disparity in drought-dust relationships in the contiguous US (CONUS) where the SEUS shows a decrease in surface dust concentrations during drought in contrast to the expected increase in dust found in other CONUS regions. Surface fine dust was found to decrease by ~0.23 $\mu g/m^3$ with a unit decrease of SPEI in the SEUS, as opposed to an increase of ~0.12 $\mu g/m^3$ in the west. The anomalies of dust elemental ratios, satellite aerosol optical depth (AOD), and dust extinction coefficients suggest that both the emissions and trans-Atlantic transport of African dust are weakened when the SEUS is under droughts. Through the teleconnection patterns of negative North Atlantic Oscillation (NAO), a lower than normal and more northeastward displacement of the Bermuda High (BH) is present during SEUS droughts which results in less dust being transported into the SEUS. At the same time, enhanced precipitation in the Sahel associated with the northward shift of the Intertropical Convergence Zone (ITCZ) leads to lower dust emissions therein. Of the ten selected models participating in the sixth phase of the Coupled Model Intercomparison Project (CMIP6), GISS-E2-1-G was found to perform the best in capturing the drought-dust sensitivity in the SEUS. This study reveals the mechanism of how droughts influence aerosol abundance through changing long-range transport of dust.

## 1 Introduction

Mineral dust plays an important role in the climate system by modifying the Earth's energy budget through direct aerosol-radiation forcing and indirect aerosol-cloud interactions (Tegen et al., 1996; Sassen, 2002; Carslaw et al., 2010). Fine mode mineral dust with an aerodynamic diameter of less than 2.5 µm can be transported over long distances and has a wide-ranging socioeconomic effect such as degeneration of air quality, disruption of public transport by poor visibility, and reduction of soil productivity (Middleton, 2017). Dust events can also be linked with a higher risk of valley fever and other respiratory and cardiovascular diseases (Karanasiou et al., 2012; Tong et al., 2017), and more non-accidental mortality (Crooks et al., 2016). Lifted by strong winds from arid and bare land, dust particles in the atmosphere are significantly modulated by hydroclimate variables, such as precipitation, temperature, relative humidity, and soil moisture (Achakulwisut et al., 2017; Brey et al., 2020; Pu and Ginoux, 2018). Thus, drought, as a recurring hydroclimate extreme, can impose large changes on the abundance of dust particles in the atmosphere. As the contiguous United States (CONUS) is prone to droughts and projected to be warmer and dryer in the future (Cook et al., 2015), it is essential to quantify the drought-dust relations and evaluate the ability of climate models to capture such relations to better understand the climate-dust feedbacks.

Most of the previous studies of drought-dust sensitivity in the US focused on the southwest (Aarons et al., 2019; Achakulwisut et al., 2018, 2019; Arcusa et al., 2020; Borlina and Rennó, 2017; Kim et al., 2021) where the major dust emission sources are located (e.g. the Chihuahuan, Mojave, and Sonoran Deserts). For example, Achakulwisut et al. (2018) quantified an increase of fine dust by 0.22–0.43 µg/m$^3$ with a unit decrease of two-month Standardized Precipitation-Evapotranspiration Index (SPEI) over the US southwest across the seasons. Both observations (Aarons et al., 2019) and simulations (Kim et al., 2021) have shown that dust enhancement under droughts can be attributed to the simultaneous increase of local dust emissions and long-range transport of dust from Asia. The observed drought-dust relationship can be used as a process-level metric to evaluate dust simulation in coupled chemistry-climate models and Earth system models. For example, a recent evaluation of dust emissions in 19 models participating in the sixth phase of the Coupled Model Intercomparison Project (CMIP6) found that interannual variations of dust emissions simulated by these models are strongly correlated with drought over major dust source regions (Aryal and Evans, 2021).

While the abovementioned studies improved our understanding of dust-drought relationships in dust source areas, regions predominantly influenced by long-range transported dust such as the southeastern US (SEUS) have received less attention. The dusty Saharan air from western Africa can reach the SEUS during boreal summer through long-range transport across the tropical Atlantic Ocean and Caribbean Basin (e.g., Perry et al., 1997; Prospero et al., 2010). Fine dust is estimated to contribute to 20-30% of the total particulate matter smaller than 2.5 µm (PM$_{2.5}$) aerodynamic diameter at the surface in the southeast during summertime (Hand et al., 2017). Extreme "Godzilla" dust events have occurred in recent years, leading to considerably worse air quality in the southeast region (Yu et al., 2021). In our previous study, Wang et al. (2017) estimated that growing-season (March-October) droughts during 1990-2014 caused an average fine dust increase of 27% in the west and 16% in the Great Plains, with a much lower effect on fine dust

in the southeastern and northeastern US. That study used a coarse time scale (i.e., averaging of the eight-month
growing season) which may not fully capture the episodic nature of dust emissions or dust transport.
Here we improve upon previous studies by using drought and dust datasets of better spatial coverage and finer
temporal scales (Section 2). In Section 3.1, we first examine how the spatial distributions of surface fine dust change
with weekly and monthly drought indices over the CONUS. The finer-scale analysis unmasks spatial disparity in
drought-dust relationships where the SEUS stands out from the rest of CONUS in that it shows a decrease in surface
dust concentrations during drought in contrast to the expected increase in dust found in other regions. We then focus
on the southeast, an area largely overlooked by prior studies of dust response to drought, and investigate in Section
3.2 how drought conditions in the SEUS affect the trans-Atlantic transport of African dust.
Among the surface dust measurement datasets examined in this study, the Barbados site located in the eastmost of the
Caribbean Windward Islands is the only long-term site on the main outflow pathway of African dust to the SEUS,
which is suitable to evaluate dust-drought relationships simulated by coupled climate-chemistry models. The surface
dust mass concentration has been continuously measured at the Barbados site since August 1965. This rare and unique
dataset was widely used to improve our understanding of the variations of African dust transport and model evaluations
(Chiapello et al., 2005; Prospero and Nees, 1986; Zuidema et al., 2019). Given the correct sensitivity of dust emissions
to drought in CMIP6 models (Aryal & Evans, 2021), in Section 3.3 we use the dust-drought relationship at the
Barbados site to evaluate the performance of ten CMIP6 models in capturing the drought-dust sensitivity in the SEUS.
**2 Data and Methods**
The datasets and related variables used in this study were summarized in Table S1-2 with details given below.
**2.1 Drought indicator**
The US Drought Monitor (USDM) index was selected as the primary drought indicator because it incorporates not
only objective indicators but also inputs from regional and local experts around the country (Svoboda et al., 2002).
USDM maps have been released every week from 2000 to the present on its website (https://droughtmonitor.unl.edu/).
There are five dryness categories on the map, labeled Abnormally Dry (D0), Moderate (D1), Severe (D2), Extreme
(D3), and Exceptional (D4) Drought. We converted these maps into $0.5° \times 0.5°$ gridded data and combined D2-D4
levels as "severe drought" due to limited data availability caused by their low spatial coverage if treated individually
(Li et al., 2022). Non-drought (wet and normal) conditions, denoted as N0, are defined when a grid is not under any
of the five dryness categories. There are 262 weeks in total during our study period from 2000 to 2019 summers (June,
July, August; JJA). To compensate for the categorical nature of the USDM data, one-month gridded SPEI data from
the global SPEI database (http://sac.csic.es/spei/) with a spatial resolution of $0.5° \times 0.5°$ and a temporal range of 1973-
2018 was also used to conduct statistical analysis (e.g., correlation and regression). The criteria of SPEI < -1.3 and
SPEI > -0.5 were applied to denote severe drought and non-drought conditions, respectively, as suggested by Wang
et al. (2017).

## 2.2 Surface dust and satellite products

To expand the spatial coverage, we created a gridded daily fine dust dataset (0.5° × 0.5°) that aggregates site-based observations from the Interagency Monitoring of Protected Visual Environments (IMPROVE) network using the modified inverse distance weighting method as done by Schnell et al., (2014). Fine dust data from the IMPROVE sites has been widely used by previous studies to investigate surface fine dust variations (Achakulwisut et al., 2017; Hand et al., 2017; Kim et al., 2021). US Environmental Protection Agency Chemical Speciation Network (EPA-CSN) also provides long-term dust data, but the CSN sites are located primarily in suburban and urban areas, hence including extreme values from urban environments which may confound the drought signals. In addition, CSN network uses different sampling practices and analytical methods from IMPROVE which can lead to systematic differences in dust measurements (Hand et al., 2012b; Gorham et al., 2021). Thus, we only used IMPROVE dataset in this study. To reduce the artifact caused by different data completeness (e.g., old sites retired and new sites started), we selected the sites with data records longer than 5 years during the study period for interpolation (Figure S1). We used the latest version of total surface dust data at the Barbados site (Figure 5a) created and published by Zuidema et al. (2019). The Barbados JJA monthly data was averaged from at least 20 daily samples in each month between 1973 and 2014.

We combined Level3 daily aerosol optical depth AOD (550nm) retrieved from Moderate Resolution Imaging Spectroradiometer (MODIS) aboard Aqua (MYD07_D3 v6.1) and Terra (MOD08_D3 v6.1) with a resolution of 1° × 1° from 2003 to 2019 (Payra et al., 2021; Pu and Jin, 2021) to examine the westward transport of African dust. Level3 monthly cloud-free dust extinction coefficients at 532nm between 2006 and 2019 from Cloud-Aerosol Lidar and Infrared Pathfinder Satellite Observation (CALIPSO) satellite were also used to analyze the vertical profiles of trans-Atlantic dust plumes. The CALIPSO data was obtained from https://asdc.larc.nasa.gov/project/CALIPSO with a 2° × 5° horizontal grid and a vertical resolution of 60 m up to 12km from the ground.

## 2.3 Meteorological data

To analyze the emission and transport of African dust, several meteorological variables were applied. Daily precipitation was taken from the Global Precipitation Climatology Project version 1.3 (GPCP V1.3). The data is a satellite-based global product from 1996 to the present with a 1° x 1° spatial resolution. Other variables, including zonal (U) and meridional (V) winds, and geopotential height at different pressure levels were from the European Centre for Medium Range Weather Forecast (ECMWF) reanalysis version5 (ERA5) dataset. Weekly data was averaged from hourly data with a resolution of 0.25° x 0.25°. Monthly North Atlantic Oscillation (NAO) data was obtained from the Climate Research Unit (CRU) calculated as the difference of normalized sea-level pressure between the Azores and Iceland (Jones et al., 1997).

## 2.4 CMIP6 AerChemMIP models

Ten models from the CMIP6 Aerosol Chemistry Model Intercomparison Project (AerChemMIP) were selected: BCC-ESM1, CESM2-WACCM, CNRM-ESM2-1, EC-Earth3-AerChem, GFDL-ESM4, GISS-E2-1-G, MIROC6, MRI-ESM2-0, NorESM2-LM, and UKESM1-0-LL. They are the only models found by the time of writing with  dust mass

ratio outputs from historical simulations with prescribed sea surface temperature in the AerChemMIP project.
NorESM2-LM is the only model containing ensembles (two members) and the ensemble mean was used here. All the
model outputs cover the period from 1850 to 2014. Dust emissions are interactively calculated based on factors such
as surface wind speed, soil type, and aridity. Dust particles are resolved to different size bins ranging from 0.01 to 63
μm in diameter. More information and references (Dunne et al., 2020; Kelley et al., 2020; Séférian et al., 2019;
Yukimoto et al., 2019; Wu et al., 2020; Danabasoglu et al., 2020; van Noije et al., 2021; Tatebe et al., 2019; Seland
et al., 2020; Senior et al., 2020) for each model are listed in Table S2.

## 3 Results

### 3.1 Reduced dust in the southeast under droughts

Figure 1a shows the mean summertime (JJA 2000 – 2019) surface fine dust concentrations under non-drought
conditions (N0) and their changes under severe droughts (D2-D4) relative to non-drought. Higher concentrations (~2
μg/m$^3$) can be found in the southwest and southeast regions under non-drought conditions, reflecting the average
spatial distributions of summertime dust. Under severe droughts, most of the grids/sites display an enhanced dust
level, with the highest enhancement (~1.5 μg/m$^3$) occurring near the source regions in the southwest (e.g., Arizona
and New Mexico). This indicates higher local dust emissions under droughts, which can be attributable to regional
precipitation, bareness, wind speed, and soil moisture anomalies (Achakulwisut et al., 2017; Kim et al., 2021; Pu and
Ginoux, 2018). By contrast, reduced fine dust is shown in the southeastern grids/sites under severe drought, especially
for the ones near the coast. Density plots in Figure 1b illustrate that the overall gridded dust distributions under severe
droughts across the CONUS move towards the high end compared with non-drought conditions, with an increase of
the mode and mean value by ~0.14 μg/m$^3$ (26%) and ~0.21 μg/m$^3$ (27%), respectively. Conversely, dust distributions
over the southeast (25°-33°N, 100°-75°W; black box in Figure 1a) move to the low end with a respective decrease of
the mode and mean value by ~0.26 μg/m$^3$ (18%) and ~0.16 μg/m$^3$ (11%). Here the southeast region is delimited to
cover most of the grids/sites with negative changes in dust during drought. Expanding the region's boundary northward
will dampen the reduced dust signal or even change it to an increase (Figure S2) due to the weakened impact of African
dust on the northern US (Aldhaif et al., 2020). To test whether the spatial interpolation process could potentially cause
biases due to the low site numbers over the southeast region, Figure 1b also plots the density distribution using on-
site IMPROVE data. Similar distributions can be seen between the gridded and on-site data, except that the latter
shows a "fatter" (more variable) distribution. This indicates that the interpolation did not significantly affect the
results. We also reproduced the above analysis using SPEI-based monthly drought criteria and similar results were
found (Figure S3), except for a smaller magnitude of dust reduction in the SEUS. This indicates the weekly data can
better capture the reduced dust signal than monthly data because of the episodical nature of the African dust transport,
which typically takes about ten days to reach the SEUS (Chen et al., 2018; Pu and Jin, 2021).

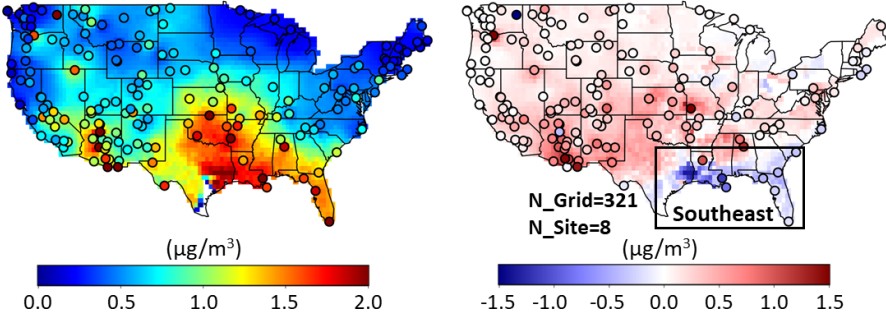

(a) Dust distribution under non-drought conditions (left) and its changes from severe drought conditions (right)

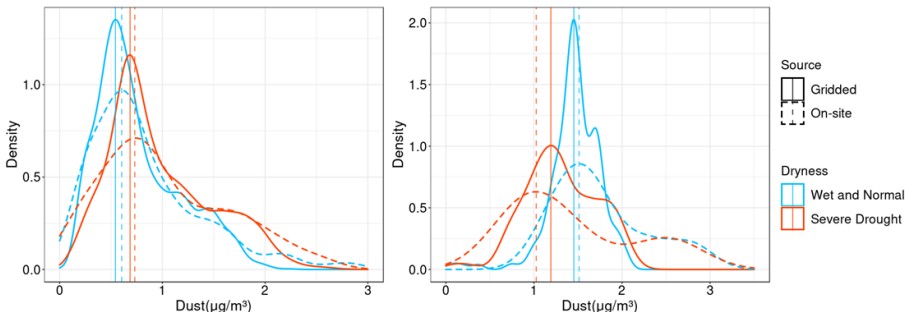

(b) Dust density plot over the CONUS (left) and the southeast region (right)


**Figure 1. (a) Maps of the mean gridded and in-situ (dots) fine dust under USDM-based non-drought (wet and normal) conditions (left) from 2000 to 2019 JJA and its changes from severe drought conditions (right). The number of grids and sites within the southeast region is denoted by N_Grid and N_Site, respectively. (b) Comparisons of density distributions of gridded (solid lines) and in-situ (dash lines) fine dust concentrations during 2000-2019 JJA under drought (red lines) and non-drought (blue lines) conditions over the CONUS (left) and southeast region (right), respectively. Vertical dash and solid lines indicate the modes.**

To further quantify the drought-dust relationship, we conducted a linear regression between SPEI and dust concentrations, taking advantage of the non-categorical nature of SPEI. The slopes of the regression at each grid are shown in Figure 2a. Almost all the grids in the western CONUS have significant negative slopes at a 95% confidence level. As negative SPEI values indicate drought, these negative slopes reveal an increasing level of dust with dryer conditions. The highest value about 0.6 $\mu g/m^3$ per unit decrease of SPEI occurs in Arizona, which is also indicative of higher dust emissions under drought consistent with the composite analysis in Figure 1. However, not all the grids in the southeast exhibit significant positive slopes as expected from Figure 1. This may imply a non-linear relationship that cannot be identified via composite analysis. To better explain this, we compared the changes in regional mean dust concentrations with SPEI bins between the southeast (as defined in Figure 1) and west (100°W westwards) in Figure 2b. We first calculated the average dust concentration by grid for each SPEI bin and then averaged grid-mean dust per SPEI bin to get the regional-mean dust concentration. The SPEI bins were selected so that the number of grids at each SPEI bin is greater than 160 (~50% out of 321 grids) over the SEUS to ensure a good regional coverage. As shown in Figure 2b, the regional-mean approach reveals a clear nonlinear pattern for the southeast with dust decreasing as the absolute value of SPEI increases in both wet (SPEI > 0.5) and dry (SPEI < 0) portions. By contrast, the west exhibits a linear relationship throughout the SPEI range. While both regions are consistent under non-drought conditions (SPEI > 0) where dust concentrations decrease with increasing wetness due to increased washout, they

diverge under drought conditions (SPEI < 0). In the western US, dust concentrations follow the expected pattern of
being higher with increasing dryness because of the dominance of local dust emissions, which are linearly related to
aridity (Duniway et al., 2019). To capture the nonlinear relationship in the SEUS, we conducted the linear regression
using only the lowest six SPEI bins under dry conditions (SPEI < 0.5). The resulting regression slope is 0.23 $\mu g/m^3$
per unit of SPEI for the southeast and -0.12 $\mu g/m^3$ per unit of SPEI for the west, respectively. In light of the regional-
mean analysis, we recalculated the slopes at each grid under drought conditions only (SPEI <0) in Figure 2c. Compare
to Figure 2a, more grids in the SEUS show a positive slope between surface dust and SPEI while the negative slope
still dominates in the rest of CONUS. Most grids with statistically significant positive slopes are found near the coast
(e.g., southern Texas and Louisiana). As SPEI is more negative with increasing dryness, the positive slope in the
southeast means a decrease of dust with increasing dryness which is consistent with the result from Figure 1 based on
USDM. Hereafter we focused on the southeast region and investigated why surface fine dust in this region shows an
opposite response to droughts compared with other CONUS regions.

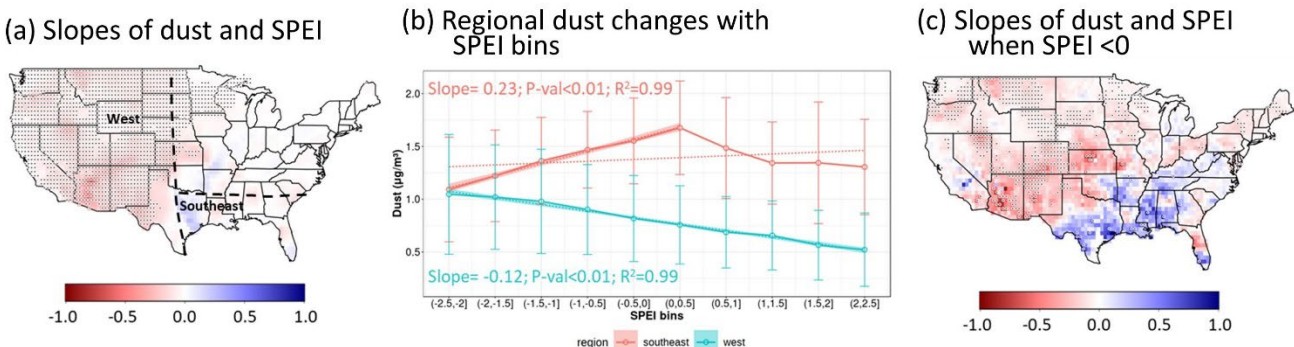


**Figure 2. (a) Maps of the linear regression slopes between fine dust concentrations and SPEI during 2000-2018 JJA. Black**
**dots denote the grids with regression significance at a 95% confidence level. Dash lines mark the boundaries of the west**
**and southeast regions. (b) Regional average dust varies with SPEI bins over the west and southeast with error bars**
**indicating one standard deviation. Dash lines display linear regression results with shadings showing the 95% confidence**
**level. The numbers indicate the slopes, P-values (P-val), and determination coefficient ($R^2$) of the regression using all the**
**SPEI bins in the west and only the first six bins in the southeast. (c) Same as a but using data under drought conditions**
**(SPEI<0) only.**
Dust elemental ratios contain important information signifying the dust particle origins (e.g., local or transport).
African dust, relative to Asian and local dust, normally has higher Fe:Ca (> 1.50) and Al:Ca (> 2.60) ratios, and lower
K:Fe (< 1.10) and Si:Al (< 2.90) ratios (Aldhaif et al., 2020; Gonzalez et al., 2021; VanCuren and Cahill, 2002). Based
on these reported thresholds, we analyzed dust elemental observations at eight sites within the southeast region (Figure
1a) and compared how the elemental ratios changed under severe drought based on the USDM drought indicator. The
results are displayed in Figure 3, with more statistical descriptions listed in Table S3. Under non-drought conditions
(wet and normal), the ratios are generally within the typical ranges mentioned above, indicating the dominance of
African dust over Asian dust and locally-emitted dust as reported by other studies (Aldhaif et al., 2020; VanCuren and
Cahill, 2002). Under severe drought, Fe:Ca and Al:Ca become lower; K:Fe and Si:Al become higher. All these changes
are in the direction of reducing the characteristic elemental ratios of African dust. Most of the Fe:Ca, Al:Ca, and K:Fe
ratios under severe drought have their medians falling below the reported thresholds of African dust. This indicates a
significantly reduced dust source from Africa. As dust deposition is unlikely to increase under drought conditions, the
lower signature of African dust in surface dust under severe drought is most likely attributable to the reduced import
of African dust to the SEUS, which is discussed below.

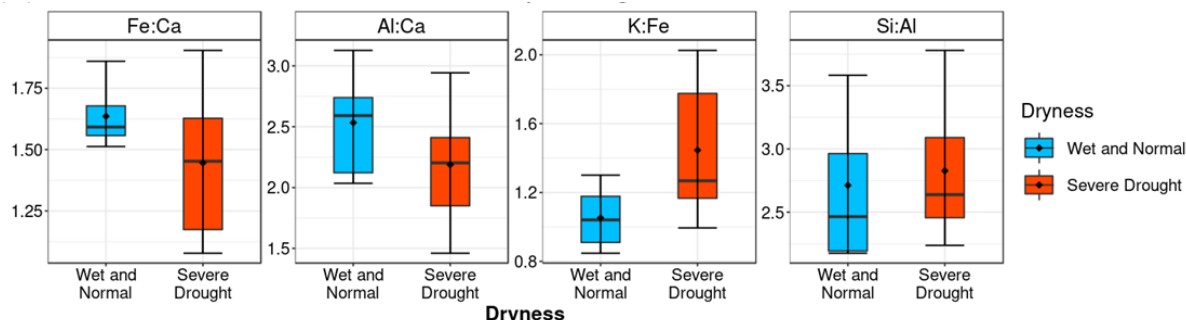

**Figure 3. Boxplots of four dust elemental ratios under non-drought (wet and normal) and severe drought conditions.**
**Observations are from eight IMPROVE sites in the southeast region shown in Figure 1a. The upper and lower whiskers of**
**the boxplots represent the ninth and first quantile, respectively. Black dots indicate the mean values. Detailed values of this**
**figure can be found in Table S3.**

### 3.2 Weakened trans-Atlantic dust transport under droughts

In this section, we examined how the trans-Atlantic transport of African dust changes with droughts in the southeast.
To do so, we first selected regional-scale drought events to better depict the aridness across the southeast, and then
associated these events with the long-range transport of African dust and compared them with regional-scale non-
drought events. On a weekly scale (USDM-based), we first examined the percentage of grids covered by D2-D4
droughts over the SEUS in an increasing order (Figure S4a). There appears to be a 'turning point' at around 30%, after
which the percentage increases much faster, suggesting a regional expansion of severe drought. Therefore, we selected
regional severe drought events based on the threshold of more than 30% of the southeastern grids under D2-D4
droughts. Figure S4a also shows that the percentages of grids under N0 or D0-D1 fall between 30% and 60% in most
of the weeks and they can be quite close (e.g., 50% under N0 and 47% under D0-D1) in some weeks. To exclude such
weeks from non-drought conditions and reduce the impact of mild drought (D0-D1), we set the threshold of regional
non-drought events as more than 70% of the southeastern grids under N0. To select regional severe drought events on
a monthly scale (SPEI-based), we used the threshold of the lowest 20% quantile of regional-mean SPEI since the
criteria of 30% of the grids under D2-D4 is nearly at the top 20% quantile of all the weeks. Months with regional-
mean SPEI greater than the top 20% quantile are considered as non-drought events. We tested other thresholds for
selecting severe droughts and non-droughts events and found consistent results in the difference of dust under severe
drought relative to non-drought events (Figure S4b-c), which indicates our conclusions are not sensitive to the
selection of these thresholds. The time series in Figure 4 shows that the regional severe drought events mainly occurred
in 2000, 2006, 2007, and 2011 JJA.

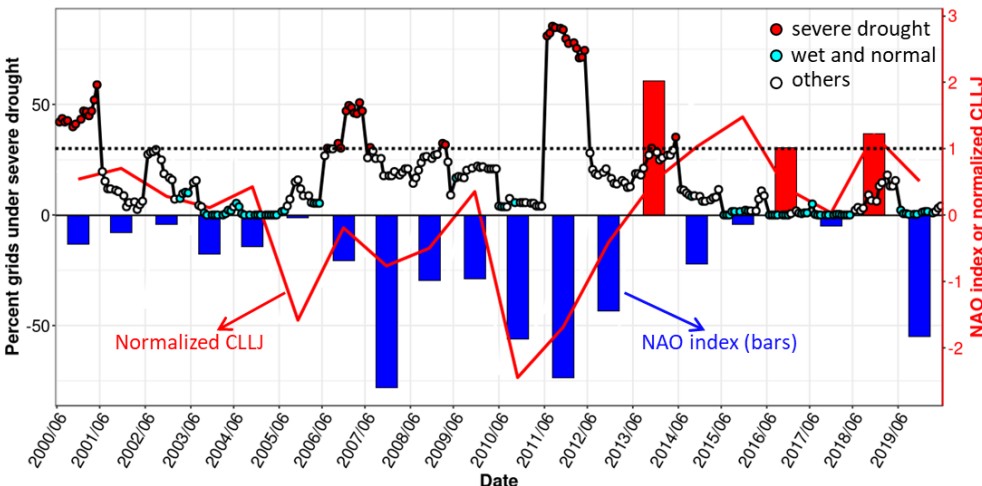


**Figure 4. Time series of weekly regional dryness levels indicated by the percentage of grids under severe drought (D2-D4) in the southeast area (filled dots; left axis), the JJA-mean North Atlantic Oscillation (NAO) index (bars; right axis), and normalized Caribbean low-level jet (CLLJ; red line; right axis). The black dash line indicates the position of 30%.**

Based on the selected regional drought and non-drought periods in the SEUS, we compiled the composite AOD from MODIS for drought and non-drought conditions. Figure 5a displays the maps of non-drought mean AOD and the changes in AOD during severe droughts. Horizontally, the major transport pathway of the dusty African air is within 10°-20°N, 100°-0°W (red box), as indicated by the higher AOD values than its surroundings. The dust flow, emitted from northern Africa (e.g., Sahara Desert and Sahel), travels through the tropical Atlantic, the Caribbean Sea, and the Gulf of Mexico before reaching the SEUS. Under droughts, almost all the AOD values along that pathway show negative differences, which indicates both the African dust transport and emissions (mainly from the Sahel) are depressed when the SEUS is under droughts. In addition, the difference map presents an enhanced dust band to the north of the major transport pathway (20°N-30°N), which is indicative of the northward shift of the transport pathway. To further explore this, we compared in Figure 5b three meridional cross sections of AOD between 0 and 30°N averaged over different longitudinal portions of the transport pathway: near the source region (Section 1; 20°W-30°W), in the middle of the pathway (Section 2; 50°W-60°W), and over the Gulf of Mexico (Section 3; 85°W-95°W). Section 1 and 2 show that the peak AOD values are lower under severe droughts with their corresponding latitudes moving 2° and 1° northward, respectively. However, almost all the AOD values in section 3 are lower under severe drought than non-drought conditions with no such northward movement observed. This indicates the enhanced dust band between 20°N-30°N does not enter the Gulf of Mexico and reach the SEUS, hence not offsetting the reduced dust in the SEUS under severe drought.

To better demonstrate the dust changes along the major transport pathway, we also examined the vertical profiles of the dust extinction coefficient from CALIPSO along the pathway (Figure 5c). Since the CALIPSO data is monthly, we used the SPEI-based drought events defined above. The dust particles can be injected up to ~4km altitude from the source region through strong desert surface heating (Alamirew et al., 2018; Flamant et al., 2007), low-level wind convergence (Bou Karam et al., 2008), synoptic-scale disturbance (Knippertz and Todd, 2010) and other processes

(Francis et al., 2020), and then descend to lower levels as they travel westwards. Such vertical structures have been
discerned by previous studies (Prospero and Mayol-Bracero, 2013; Ridley et al., 2012). Similar to Figure 5a, a
decreased dust extinction coefficient is found along the vertical transport pathway, which verifies the conclusion that
both the transport and emissions of African dust are weakened when the SEUS is under droughts.

(a) MODIS AOD under wet and normal conditions (left) and its changes from severe drought conditions (right)

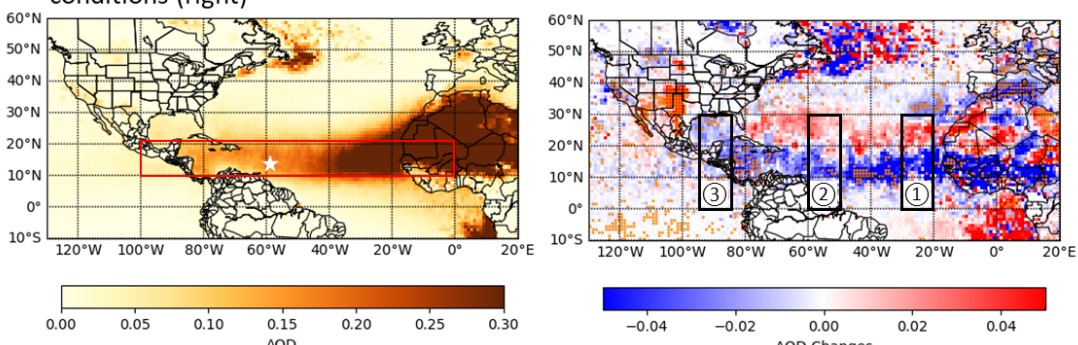

(b) Three meridional cross sections (0-30⁰N) of AOD under non-drought (blue) and severe drought (red) conditions.

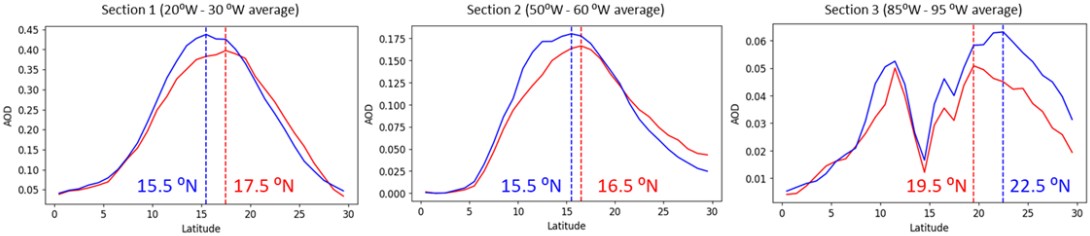

(c) Dust extinction coefficient vertical profile under wet and normal conditions (left) and its changes from severe drought conditions (right)

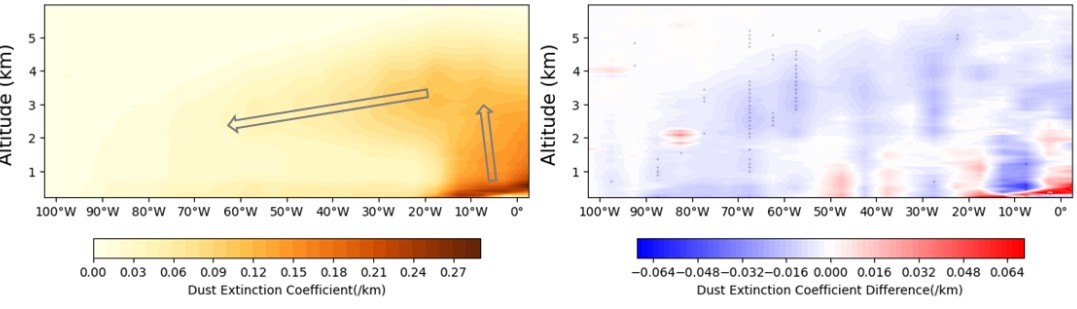


**Figure 5. (a) Maps of AOD (550 nm) under non-drought (wet and normal) conditions (left column) and its changes during**
**severe droughts (right column). The severe drought and non-drought periods were chosen based on the weekly time series**
**shown in Figure 4. The white asterisk denotes the location of the Barbados site (13°6'N, 59°37'W). Black and red rectangles**
**denote the locations of the cross sections in b and c, respectively. (b) Meridional cross sections between 0-30°N averaged**
**near the source region (section 1; 20°W-30°W), in the middle of the transport pathway (section 2; 50°W-60°W), and over**
**the Gulf of Mexico (section 3; 85°W-95°W) under non-drought (blue) and severe drought (red) conditions. The dash lines**
**and associated numbers indicate the latitudes with the maximum values of AOD. These three sections correspond to the**
**black rectangles labeled in the right panel of 5a to show their locations. (c) Mean vertical profiles of dust extinction**
**coefficient during non-drought (left) and severe drought (right) periods across the major transport pathway (red rectangle**
**in a). The severe drought and non-drought periods were chosen based on monthly SPEI between 2006 and 2018. Black or**
**orange dots in a and c (right column) indicate the significant difference at a 95% confidence level relative to non-drought**
**conditions.**

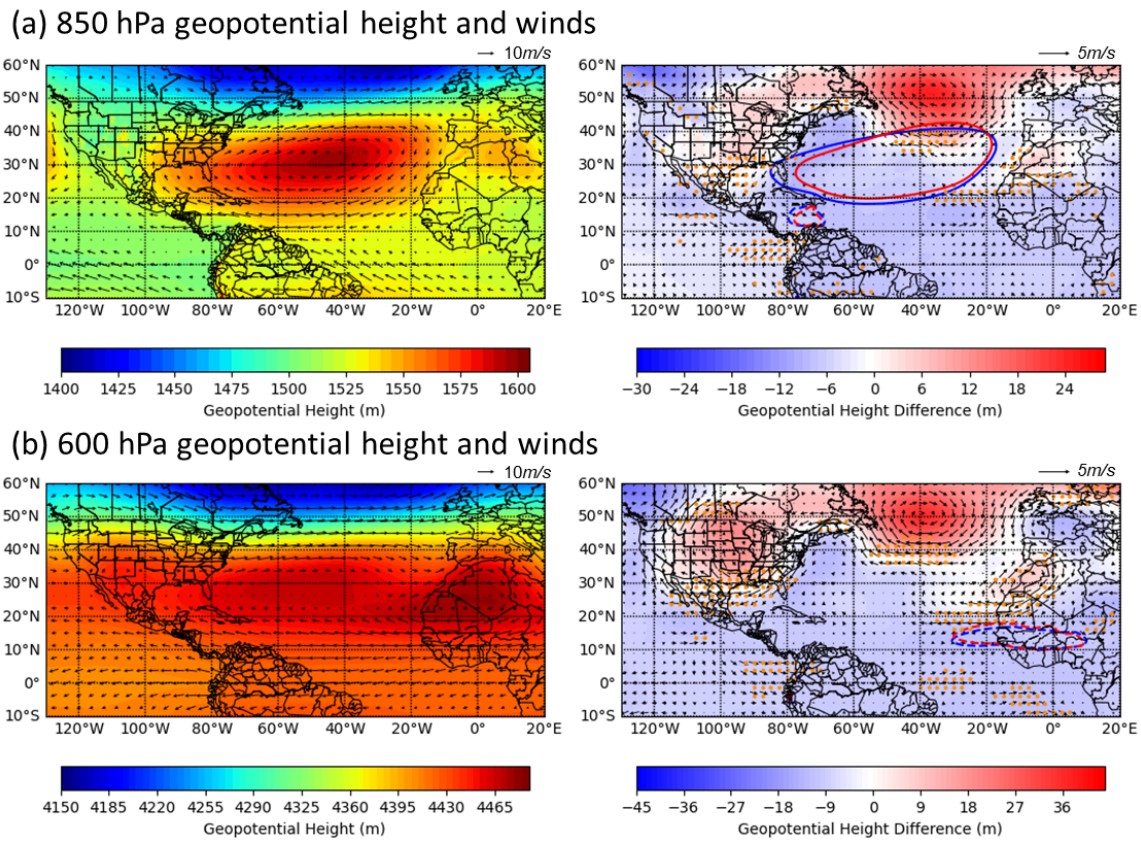

**(a) 850 hPa geopotential height and winds**

**(b) 600 hPa geopotential height and winds**

**Figure 6. Maps of geopotential height (shadings) and wind vectors (arrows) at 850 hPa (a) and 600 hPa (b) under the USDM-based SEUS regional non-drought (wet and normal) conditions (left column) and their changes during severe drought periods (right column) from 2000 to 2019 JJA. Solid lines in a indicate the edge of Bermuda High under non-drought (blue) and severe droughts (red). Dash lines show the edge of Caribbean low-level jet (a) and African easterly jet (b) under non-drought (blue) and severe droughts (red). Orange dots (right column) indicate the grids with significant differences of zonal winds at a 95% confidence level.**

The teleconnections between the SEUS droughts and the transport and emissions of African dust are displayed in Figure 6. At low levels near the central North Atlantic, a semipermanent high-pressure system called North Atlantic Subtropical High (NASH) or Bermuda High (BH) favors the dust transport with its southwestward extensions towards the Caribbean and the Gulf of Mexico steering dust into CONUS (Doherty et al., 2008; Kelly and Mapes, 2011). This can be clearly seen from the anticyclonic wind circulations in Figure 6a. Using the 1560m contour (solid lines in Figure 6a) as the edge of the BH following Li et al. (2011), a retreat of the BH towards the northeast can be recognized under droughts, causing northerly wind anomalies over the Caribbean and the Gulf of Mexico. As the normal winds are southerly, the northerly wind anomalies result in a weakened dust transport into the SEUS. Such wind anomalies can also prevent the enhanced dust band (Figure 5a) from entering the SEUS. Accompanied by the southwestward extension of BH, the Caribbean low-level jet (CLLJ), defined as the mean zonal wind speed at 925 hPa over 11°–17°N, 70°–80°W, is also used to assess the westward transport of dust over the Caribbean Sea (Wang, 2007). The edge of CLLJ is denoted by the 12 m/s zonal wind speed contour (dash lines in Figure 6a). The shrinkage of CLLJ under droughts further verifies the weakened dust transport at low levels.

The geopotential height pattern associated with these circulation and jet changes is a higher than normal subpolar low
and lower than normal BH, which is consistent with the negative phase of North Atlantic Oscillation (NAO) (Barnston
and Livezey, 1987). A negative phase of NAO has been proven to be teleconnected with dry weather over the SEUS
and northern Europe, and wet weather over southern Europe and the Mediterranean due to fewer and weaker storms
caused by the reduced pressure gradient between the subtropical high and low (Hurrell, 1995; Visbeck et al., 2001).
The time series in Figure 4 show severe drought events (e.g., 2011) are associated with strong negative NAO and
abnormally low CLLJ. Similarly, we found both NAO and CLLJ are positively correlated with SPEI over the SEUS
(Figure 7a, c) with their corresponding mean magnitude reduced by 0.80 and 1.27 m/s, respectively, compared with
non-drought conditions (Figure 7b, d). This further confirms the weakened low-level dust transport into the southeast
region. It is also noted in Figure 4 that in some years (e.g., 2000 and 2006) the severe drought is not closely associated
with strong negative NAO. The reason is that other processes, such as El Niño and the Southern Oscillation (ENSO)
and Pacific Decadal Oscillation (PDO), can also trigger drought conditions over the SEUS (Piechota and Dracup,
1996; Cook et al., 2007; Pu et al., 2016). For example, the cold phase of ENSO, known as La Niña, is linked with the
fast-developing droughts over the SEUS in 2000 and 2006 by Chen et al. (2019) despite the NAO index was not too
strong in those years. Although many factors contribute to the SEUS droughts, the abnormal circulation patterns
related to the negative phase of NAO impose more influence on the African dust transport, and thus we focus on NAO
in this study.

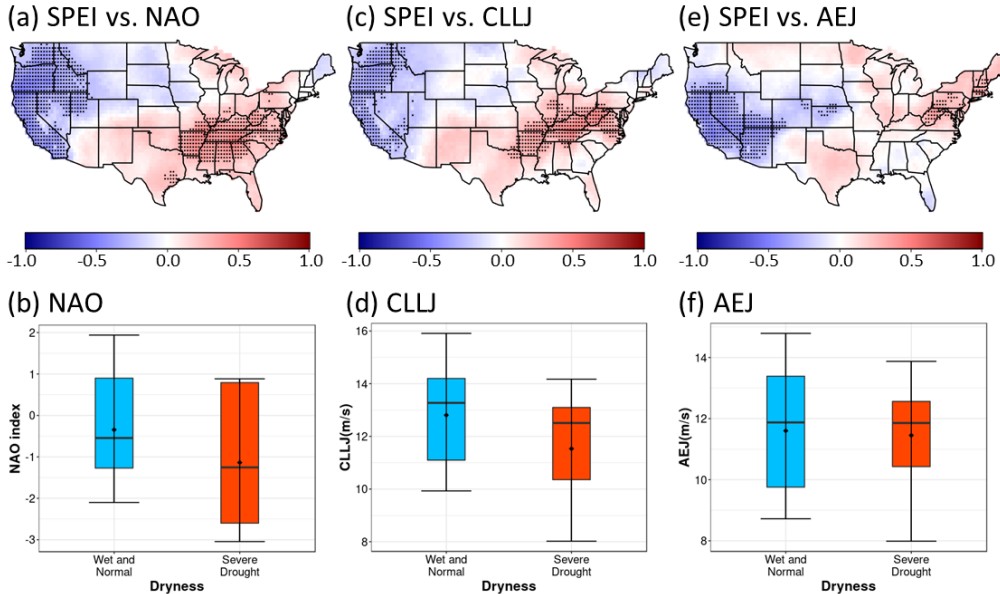


**Figure 7. Map of the correlation coefficient between SPEI and NAO (a), CLLJ (c), and AEJ (e) during 2000-2018 JJA with**
**black dots denoting the significant correlation at a 95% confidence level. And the boxplots of NAO (b), CLLJ (d), and AEJ**
**(f) distributions under non-drought (wet and normal) and severe drought conditions.**
The westward dust propagation at high levels (e.g., at ~3 km altitude) mainly occurs near the source region after being
injected from the surface (Figure 5c). The African easterly jet (AEJ), defined as the average zonal wind speed at 600
hPa over the area of 10°–15°N, 30°W–10°E (Cook, 1999), has been widely linked with the transport of the African
dust towards tropical Atlantic (e.g., Jones et al., 2003; Pu & Jin, 2021). Another strengthened high pressure over North
Africa (Saharan Anticyclone) at 600 hPa (also seen at 850 hPa) leads to stronger winds to the northern rim of AEJ
(Figure 6b). However, the core jet area seems to be less affected as shown by the comparable magnitude of AEJ
between non-drought and drought conditions in Figure 7f. The edge of AEJ, denoted by the 11 m/s zonal wind contour
(dash lines in Figure 6b), only slightly moves northwards and does not show noticeable expansion or shrinkage. There
are no significant correlations between SPEI and AEJ over the SEUS either (Figure 7e), which indicates weak
teleconnection between droughts in the SEUS and the dust transport strength at a high level. The abnormally high
Saharan Anticyclone at both 850 hPa and 600 hPa (Figure 6a-b) is likely to increase both emissions and transport of
dust from the Sahara Desert, thus causing the enhanced dust band (20°N-30°N) in Figure 5a.

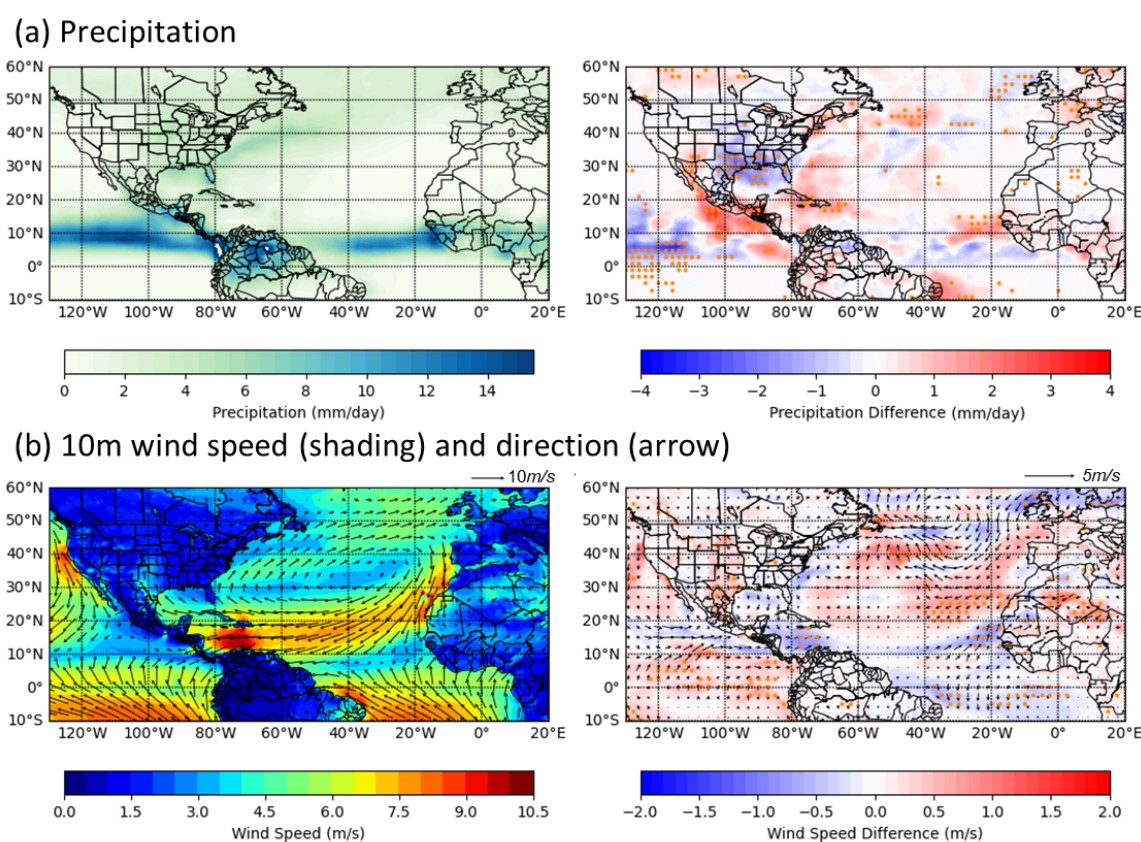


**Figure 8. Maps of precipitation (a) and 10m wind speed (shadings in b) and directions (arrows in b) under the USDM-based**
**SEUS regional non-drought (wet and normal) conditions (left column) and their changes during severe drought periods**
**(right column) from 2000 to 2019 JJA. Orange dots (right column) indicate the grids with significant differences of**
**precipitation (a) and wind speed (b) at a 95% confidence level.**

Precipitation is one of the dominant factors influencing African dust emissions (Moulin and Chiapello, 2004). A
maximum precipitation zonal belt near 5°–10°N can be seen under non-drought conditions in Figure 8a, which
represents the location of the Intertropical Convergence Zone (ITCZ). We found enhanced precipitation in southern
West Africa (10°–20°N, 30°–0°W) and the Caribbean Sea, which will reduce dust emissions from the major source
region of Sahel (e.g., southern Mauritania and Mali) and enhance the wet scavenging of dust to the Caribbean Sea. A

significant anticorrelation between summertime Sahel precipitation and NAO has been reported by previous studies on a multidecadal scale (Folland et al., 2009; Linderholm et al., 2009), which is caused by the northward displacement of ITCZ shifting the "rain belt" into the Sahel region in response to a warmer North Atlantic (Sheen et al., 2017; Yuan et al., 2018). By locating the maximum rainfall within 0°–20°N, 30°–0°W following Liu et al., (2020), we found an average of ~0.6° norward movement of ITCZ during the SEUS droughts. This can also be seen from the southwesterly 10m wind anomalies over the same region, which are contradicting to the northeasterly winds under non-drought conditions (Figure 8b). Surface wind speed is another important factor associated with dust emissions in this region (Evan et al., 2016). However, Figure 8b does not show clear negative anomalies over the Sahel region under droughts, which implies that surface wind speed is not a significant factor causing the weakened dust emissions in the Sahel. Instead, stronger winds are found over part of the Sahara (20°–30°N, 5°W–10°E), which would increase the dust emissions therein and contribute to the enhanced dust band displayed in Figure 5a.

In summary, the reduction of surface fine dust in the SEUS under severe drought results from the weakened African dust transport and emissions from the Sahel through the teleconnection patterns of negative NAO. The weaker and less southwestward extension of the BH reduces the wind speed over the Caribbean and the Gulf of Mexico, making it less favorable for African dust to enter the SEUS at low levels. Intensified precipitation over the Sahel related to the northward shift of ITCZ is the main factor causing lower Sahelian dust emissions during the SEUS droughts, and this factor dominates over surface wind speed changes.

### 3.3 CMIP6 model evaluation

In this section, we evaluated the surface dust concentrations from ten CMIP6 models regarding their capability of capturing the drought-dust relationships in the SEUS in comparison with the monthly observations (1973-2014; JJA) at the Barbados site. Dust values were extracted from the lowest model layer at a grid point nearest to the observation site. Out of the 120-month study period, 24 severe drought months were identified based on the same SPEI-based regional-drought criteria as described in the last section.

Figure 9a displays the scatter plots between model simulations and observations with more statistics listed in Table 1. CNRM-ESM2-1, EC-Earth3-AerChem, MIROC6, and NorESM2-LM considerably underestimate the dust concentrations by more than 16 μg/m$^3$ (70%) regardless of the drought conditions. GFDL-ESM4, MRI-ESM2-0, and UKESM-0-LL simulations have a relatively lower underestimation of ~7 μg/m$^3$ (28%), ~5 μg/m$^3$ (18%), and ~3 μg/m$^3$ (13%), respectively, with the latter being the minimum bias among all the ten models, but they do not reproduce the observed variability as indicated by the negative correlation coefficient (R) and slope. Under droughts, both the underestimations of GFDL-ESM4 and MRI-ESM2-0 are reduced by ~38% with R and slope values turning to positive or closer to zero, which indicates these two models have better performance under droughts. By contrast, the UKESM-0-LL model performs slightly worse if using drought months only, as indicated by the ~3% higher underestimation and the more negative R and slope values. An overall overestimation of ~7 μg/m$^3$ (29%), ~9 μg/m$^3$ (36%), and ~5 μg/m$^3$ (21%) was found in the simulations of BCC-ESM1, CESM2-WACCM and GISS-E2-1-G, respectively. The

negative or low R and slope values (less than 0.25) of these three models also show that they can barely capture the
dust variability. If only the drought months are considered, all three models have a better capability in predicting the
dust variability with R increasing to 0.18 (BCC-ESM1), 0.25 (CESM-WACCM), and 0.37 (GISS-E2-1-G).

(a) Comparison of monthly observations and simulations

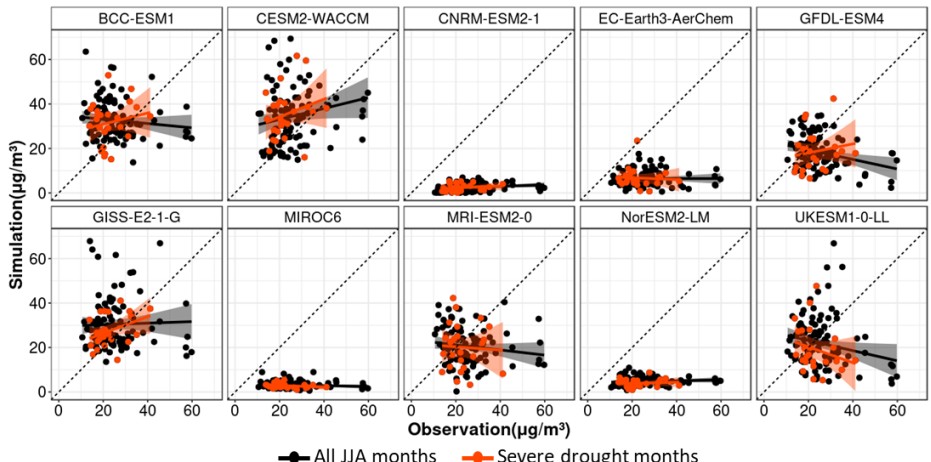

(b) Dust changes with regional-mean SPEI bins

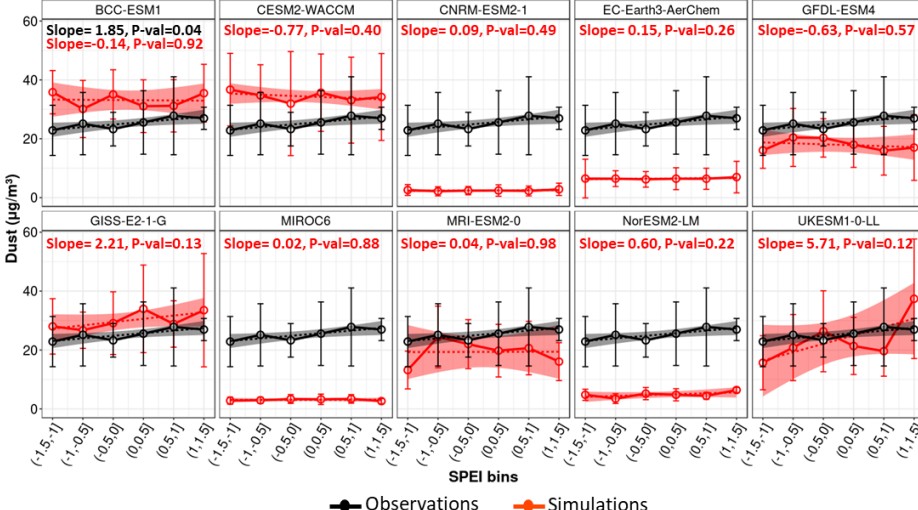


**Figure 9. (a) Scatter plots between dust observations and ten CMPI6 models during 1973-2014 JJA. Black dots and lines represent dust in all the JJA months and their linear regression fits, respectively. Red dots and lines indicate the same analysis but using the SPEI-based severe drought months only. The shadings indicate a 95% confidence level of the linear regressions. The dashed lines correspond to the 1:1 correlation. (b) Observed and simulated dust means (dots) and standard deviations (error bars) vary with the SEUS regional-mean SPEI. Dash lines represent the linear regressions of the average dust concentrations with their slopes (Slope) and P-values (P-val) listed at the top of each panel.**

The sensitivity of surface dust in response to the SEUS regional drought was also evaluated by comparing the
simulated and observed slopes of dust changes with regional mean SPEI. The results are displayed in Figure 9b.
Similar to the fine dust responses to drought in the southeast, total dust at Barbados also shows a decreasing tendency
with lower SPEI. On average, dust at the Barbados site reduces by 1.85 μg/m³ with a unit decrease of SPEI over the
southeast region. This consolidates the conclusion that the weakened across-Atlantic transport of African dust is the
reason causing the reduced fine dust in the SEUS as the Barbados site sits in the major transport pathway. UKESM-
0-LL model shows a much higher sensitivity of 5.71 µg/m$^3$ (P-value= 0.12) probably driven by the high dust value
under the wettest conditions (SPEI >1). GISS-E2-1-G simulations have a comparable sensitivity of 2.21 µg/m$^3$ (P-
value= 0.13) despite its general overestimation, which makes it outperform the other nine models with a much lower
and less statistically significant sensitivity in response to SPEI changes.
**Table 1. Evaluation metrics of ten CMIP6 models in comparison with observations at the Barbados site during 1973-2014**
**JJA. Metrics include correlation coefficient (R), mean bias (MB), normalized mean bias (NMB), root mean square error**
**(RMSE), and slope.**

| Simulations | Drought Conditions | Observed Mean (µg/m$^3$) | Simulated Mean (µg/m$^3$) | R | MB (µg/m$^3$) | NMB (%) | RMSE (µg/m$^3$) | Slope |
|---|---|---|---|---|---|---|---|---|
| BCC-ESM1 | All months | 25.19 | 32.62 | -0.11 | 7.43 | 29.49 | 15.89 | -0.09 |
|  | Severe Drought | 22.94 | 31.64 | 0.18 | 8.71 | 37.96 | 13.62 | 0.25 |
| CESM2-WACCM | All months | 25.19 | 34.32 | 0.17 | 9.13 | 36.25 | 18.20 | 0.24 |
|  | Severe Drought | 22.94 | 35.37 | 0.25 | 12.43 | 54.22 | 16.97 | 0.40 |
| CNRM-ESM2-1 | All months | 25.19 | 2.42 | 0.20 | -22.76 | -90.36 | 24.75 | 0.03 |
|  | Severe Drought | 22.94 | 2.41 | 0.17 | -20.53 | -89.50 | 21.63 | 0.04 |
| EC-Earth3-AerChem | All months | 25.19 | 6.43 | 0.002 | -18.76 | -74.47 | 21.53 | 0.001 |
|  | Severe Drought | 22.94 | 6.71 | -0.07 | -16.23 | -70.75 | 18.26 | -0.04 |
| GFDL-ESM4 | All months | 25.19 | 18.24 | -0.26 | -6.95 | -27.59 | 15.92 | -0.21 |
|  | Severe Drought | 22.94 | 18.60 | 0.16 | -4.34 | -18.92 | 11.18 | 0.20 |
| GISS-E2-1-G | All months | 25.19 | 30.43 | 0.03 | 5.24 | 20.79 | 16.19 | 0.03 |
|  | Severe Drought | 22.94 | 27.50 | 0.37 | 4.56 | 19.89 | 9.07 | 0.37 |
| MIROC6 | All months | 25.19 | 3.20 | -0.15 | -21.99 | -87.28 | 24.26 | -0.02 |
|  | Severe Drought | 22.94 | 2.85 | -0.26 | -20.08 | -87.58 | 21.36 | -0.04 |
| MRI-ESM2-0 | All months | 25.19 | 20.62 | -0.13 | -4.57 | -18.15 | 14.90 | -0.11 |
|  | Severe Drought | 22.94 | 20.11 | -0.05 | -2.83 | -12.33 | 12.94 | -0.07 |
| NorESM2-LM | All months | 25.19 | 4.73 | 0.10 | -20.46 | -81.21 | 22.74 | 0.02 |
|  | Severe Drought | 22.94 | 3.95 | 0.09 | -18.98 | -82.75 | 20.22 | 0.02 |
| UKESM1-0-LL | All months | 25.19 | 21.96 | -0.19 | -3.22 | -12.80 | 16.96 | -0.22 |
|  | Severe Drought | 22.94 | 19.22 | -0.24 | -3.71 | -16.17 | 14.11 | -0.35 |


In conclusion, BCC-ESM1, CESM2-WACCM and GISS-E2-1-G generally show an overestimation of surface dust,
while the other seven models exhibit an underestimation with the highest underestimation found in the CNRM-ESM2-
1, EC-Earth3-AerChem, MIROC6, and NorESM2-LM simulations. None of the ten models is capable of capturing
the dust variability using all the months. If using the drought months only, BCC-ESM1, CESM2-WACCM, GFDL-
ESM4, GISS-E2-1-G, and MRI-ESM2-0 perform better. GISS-E2-1-G can reproduce the dust-SPEI sensitivity much
better than the other nine models. It is noted that systematic bias should arise when comparing single-site observations
with grid-mean predictions, which could presumably cause the between-model diversity as they have different spatial
resolutions (Table S2). However, the dust-sensitivity evaluation should be less affected as its calculation depends
more on relative changes, instead of absolute values.
**4 Conclusions**
We found an opposite response of surface fine dust to severe droughts between the western and southeastern CONUS,
with an increase of ~0.12 $\mu g/m^3$ and a decrease of ~0.23 $\mu g/m^3$ per unit decrease of SPEI, respectively. Similar results
were reached by the USDM-based drought conditions, with an average decrease of 0.16 $\mu g/m^3$ under D2-D4 droughts
over the SEUS relative to non-drought conditions. The dust and drought relationship over the west/southwest region
has been investigated before due to its vicinity to the major dust source regions, and the increase of dust with drought
is expected. As the southeast region is strongly influenced by long-range transport of African dust in the summer, we
investigated how drought conditions in the SEUS can be linked with the trans-Atlantic transport of African dust.
The elemental ratios are indicative of the dominance of African dust in the southeast region. The tendency of these
ratios moving out of the normal range under severe droughts implies a reduced African dust input. The anomalies of
satellite AOD and dust extinction coefficients suggest that both the transport and emissions of African dust are weaker
during the southeast drought periods than non-drought periods. The composite analysis reveals that the weaker across-
Atlantic dust transport is through the teleconnection patterns of the negative NAO. During the drought periods, a lower
than normal and more northeastward displacement of the Bermuda High results in less dust being brought into the
SEUS at low levels from the Caribbean and the Gulf of Mexico by its southwestward extensions. This can also be
seen from a weaker and more shrinking CLLJ. Enhanced precipitation in the Sahel associated with the northward shift
of ITCZ leads to lower dust emissions therein.
At last, we evaluated ten CMIP6 models with surface dust outputs. CNRM-ESM2-1, EC-Earth3-AerChem, MIROC6,
and NorESM2-LM generally perform the worst with an up to 70% underestimation of the dust concentrations. GFDL-
ESM4, MRI-ESM2-0, and UKESM-0-LL underpredict the dust level by 28%, 18%, and 13%, respectively. BCC-
ESM1, CESM2-WACCM, and GISS-E2-1-G show a respective overestimation of 29%, 36%, and 21%. All ten models
fail to reproduce the dust variability using data from all the months, with BCC-ESM1, CESM2-WACCM, GFDL-
ESM4, GISS-E2-1-G, and MRI-ESM2-0 models significantly improving their performance if only the drought months
are used. GISS-E2-1-G outperforms other models in capturing the dust-SPEI sensitivity.
This study establishes how the local- or regional-scale drought conditions in the SEUS are linked with the long-range
transport and emission changes of African dust through teleconnections. It also reveals the mechanism of how droughts
influence aerosol abundance through changing long-range transport of dust. Thus, in order to better predict how the
local dust air quality will change in response to an increasing drought frequency in a warming climate (Cook et al.,
2015), climate and Earth system models not only need to represent various physical processes associated with the
entire dust cycle, but also should capture the abnormal atmospheric processes (e.g., circulation and precipitation)
related to droughts. Evaluation of these models should use observations of dust-drought relationships not only in dust
source regions but also in dust transported regions.

**Acknowledgments**

This research was supported by the NOAA's Atmospheric Chemistry, Carbon Cycle, and Climate (AC4) Program
(NA19OAR4310177). The authors acknowledge NASA for providing the MODIS AOD and CALIPSO data, EPA
and IMPROVE in making the dust observations. We thank individuals and groups for creating the USDM maps and
the SPEI dataset. The authors also thank the modeling groups participating in the CMIP6 AerChemMIP project for
making the surface dust outputs available.

**Data Availability**

The data used for this study can be downloaded through the links provided in Table S1 and Section 2.

**Competing interests**

The authors declare that they have no conflict of interest.

**Author contributions**

YW conceived the research idea. WL conducted the analysis. Both authors contributed to the preparation of the
manuscript.

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
