# Peer review of "Reduced surface fine dust under droughts over the southeastern"

_EGUsphere, 2022_

## Author Comment (AC1)

**Reply to Reviewers**

We sincerely appreciate all the reviewers for their constructive comments to improve the manuscript. Their comments are reproduced below followed by our responses in blue. The corresponding edits in the manuscript are highlighted with track changes.

**Reviewer #1:**

General Comments:

The manuscript "Reduced surface fine dust under droughts over the southeastern United States during summertime: observations and CMIP6 model simulations" by Li and Wang explores the drought-dust sensitivities over the CONUS based on observations and four AerChemMIP models. The authors found negative drought-dust sensitivities over the southeastern US as opposed to the usual positive sensitivities found over the western US. They associate these anomalous sensitivities over the southeastern US to decreased emissions and trans-Atlantic transport of African dust. The manuscript is well written, and the content is straight-forward with reasonable conclusions. I appreciate that the authors consider multiple observational products and analysis to add confidence to their findings. I recommend acceptance with minor revisions.

Specific Comments:

(1). In line 102-103, if I understand correctly, the EPA-CSN data is remapped from 1x1 (coarse) to 0.5x0.5 (fine) grids using bilinear interpolation? It is often recommended to use conservative remapping when regridding from coarse to fine resolution. This way we can avoid biases near high emission/near-source regions.

Response: Thanks for the suggestion. Because of the bias between IMPROVE and EPA-CSN sites as suggested by the other reviewer (comment on Line 97), we changed to use the IMPROVE sites only and recreated the dataset with a finer resolution (0.5x0.5). Therefore, the remapping process from coarse to fine grids was deleted in the revised manuscript.

(2). In figure 2a, does the southeast US region show significant regression slopes when the dust is constrained within SPEI<0.5 bins? I understand that it will contain a substantial amount of missing data, but I would be curious to see the spatial distribution of the slopes under dry conditions. Also, I do not see the point of p-value in Figure 2b (southeast case). I realize that there is a positive relationship between SPEI and dust concentration, but it is not significant based on regional averages. So, consider looking at the slopes at each grid box to show the spatial distribution of the SPEI-dust sensitivity with significance.

Response: As suggested, we calculated the regression slopes at each grid box under dry conditions only in Figure R1c (also new Figure 2c) and indeed more grid boxes in the SEUS emerge with positive slopes than in Figure 2a. On the regional scale, the insignificant regression slope was due to few SPEI bins. To avoid this issue, we applied a different way to do the analysis, which is to calculate the average dust concentration grid by grid for each SPEI bin and then average grid-mean dust per SPEI bin to get the regional-mean dust concentration. This grid-by-grid analysis expands the SPEI range to between -2.5 and 2.5 and increases the number of SPEI bins to 10 (Figure R1b). The number of grids at each bin is greater than 160 (~50% out of 321 grids) over the SEUS to ensure the bin separation can represent the regional conditions.

Dust in the SEUS still exhibits a non-linear trend with SPEI (Figure R1b). The new linear fit under dry conditions (SPEI < 0.5) over the SEUS shows a significant (P-val <0.01) slope of 0.23 µg/m$^3$ per unit of SPEI. This revised analysis suggests that the averaging by grids within different SPEI bins is essential to reduce noises and capture the signal of the drought-dust sensitivity in the SEUS. The discussion was added in Line184-199 of the revised manuscript with Figure R1 inserted as new Figure 2.

[Figure]

**Figure R1**: (a) Linear regression slopes between fine dust concentrations and SPEI. Black dots denote the grids with regression significance at a 95% confidence level. Dash lines mark the boundaries of the west and southeast regions. (b) Regional average dust varies with SPEI bins over the west and southeast with error bars indicating one standard deviation. Dash lines display linear regression results with shadings showing the 95% confidence level. The numbers indicate the slopes (Slope), P-values (P-val), and the determination coefficient (R$^2$) of the regression using all the SPEI bins in the west and only the first six bins in the southeast. (c) Same as a, but using the data with SPEI<0 only.

(3). From line 207-209, please elaborate on why these exact thresholds (30% or 70% etc.) are used. Consider putting appropriate references.
**Response**: We examined the percentage of grids covered by D2-D4 over the SEUS in increasing order (Figure R2a). There appears to be a 'turning point' at around 30%, after which the percentage increases much faster. This indicates a regional expansion of severe drought events. Therefore, we selected 30% as the threshold of regional severe drought events. Most of the weeks have 30%-60% of grids under N0 or D0-D1. In some weeks, the percentage of grids under N0 and D0-D1 can be quite close (e.g., 50% and 47% ). To avoid the inclusion of these weeks into the non-drought conditions and reduce the impact of mild drought (D0-D1), we decided to set 70% as the threshold of regional non-drought events. The threshold of 30% of the grids under D2-D4 is nearly at the top 20% quantile of all the weeks. To match this, we select the 20% quantile of regional-mean SPEI as the threshold for regional severe drought events on a monthly scale. Months with regional-mean SPEI greater than the 20% quantile are considered as non-drought events.

We also tested the results of using other thresholds by reproducing the differences of MODIS AOD (Figure R2b; USDM-based) and dust extinction coefficient (Figure R2c; SPEI-based) between severe drought and non-drought events the same as those in Figure 4 (new Figure 5). The signal of reduced dust under droughts is consistent, which means the threshold selection does not significantly affect the conclusions. These explanations were added in the main text between Line 233-245. Figure R2 was also included in the supplement file as Figure S4.

[Figure]

**Figure R2**: (a) Percentage of grids under severe drought (D2-D4; red dots) in increasing order and the corresponding percentage of grids under non-drought (N0; blue dots) and mild drought (D0-D1; orange dots). (b) MODIS AOD difference between severe drought and non-drought events if more than 40% grids are under D2-D4 (severe drought) and more than 60% grids are under N0 (non-drought; left column), or more than 20% grids are under D2-D4 (severe drought) and more than 80% grids are under N0 (non-drought; right column). (c) Dust extinction coefficient difference between severe drought and non-drought events if the regional-mean SPEI is greater (severe drought) and smaller (non-drought) than 10% quantile (left column) or 30% quantile (right column). Black or orange dots in b and c indicate the significant difference at a 95% confidence level.

(4). In figure 4b, why does the difference figure show 2 contrasting bands? Is it possible that the pathway shifted northward? What does the spatial map look like for severe drought conditions?

**Response**: Thanks for raising a good point. The spatial maps of AOD under non-drought, severe drought and the difference between these two are shown in Figure R3a. It is difficult to discern whether the transport pathway shifts norward through eyeballing these maps. Thus, we investigated three meridional cross sections of AOD from 0 to 30ºN averaged across the longitudes near the source region (Section 1; 20ºW-30ºW), in the middle of the pathway (Section 2; 50ºW-60ºW), and over the Gulf of Mexico (Section 3; 85ºW-95ºW) in Figure R3b. Indeed, Section 1 and 2 show that the peak AOD values are lower under severe droughts with their corresponding latitudes moving 2º and 1º northward, respectively. However, almost all the AOD values in Section 3 are lower under severe drought than non-drought conditions with no such northward movement observed. This indicates the enhanced dust band between 20ºN-30ºN does not enter the Gulf of Mexico and reach the SEUS, thus not offsetting the reduced dust in the SEUS under severe drought. The old Figure 4a was separated as an individual new Figure 4. The old Figure 4b-c and Figure R3b were combined as new Figure 5 with the associated discussion added in Line 262-271.

[Figure]

**Figure R3**: (a) Average of MODIS AOD under non-drought conditions (left column), severe drought conditions (middle column), and the difference between severe and non-drought conditions (right column). (b) Meridional cross sections between 0-30ºN averaged near the source region (section 1; 20ºW-30ºW), in the middle of the transport pathway (section 2; 50ºW-60ºW), and over the Gulf of Mexico (section 3; 85ºW-95ºW) under non-drought (blue) and severe drought (red) conditions. The dash lines and associated numbers indicate the latitudes with the maximum values of AOD. These three sections correspond to the black boxes and numbers in a (right column) to show their locations.

At low levels, the lower than normal and the northeastward retreat of Bermuda High (BH) as shown by the edge of BH in the new Figure 6a is partly responsible for the northward shift of the dust transport pathway. Such changes in BH fail to steer the enhanced dust flow into SEUS because the western flank of BH is less extended relative to non-drought conditions. Near the source region of the Sahara, the enhanced 10m winds shown in new Figure 8b are meant to increase the emissions of dust particles, which can then be transported westwards by stronger winds at the 600 hPa level near the Sahara ( new Figure 6b). These changes are opposite to those near the source region of the Sahel, thus causing two contrasting bands. These explanations were added to Line 308-309, 345-346, and 366-367 to fit into the context of the manuscript.

(5). Consider adding the number of realizations used in the CMIP6 model evaluation. I think the readers would be interested to see a model ensemble mean response as well in Figure 8 (even though there are only 4 models and probably 10 realizations).
**Response**: It is a good suggestion to add ensemble means in the evaluation. However, we did not find any ensembles of these four models under the AerChemMIP project with prescribed sea surface temperature settings (histSST). That is why we did not include ensembles initially. The Atmospheric Model Intercomparison Project (AMIP) project is reported to have more ensembles (Zhao et al., 2021). Surprisingly, six more models were found after we changed the searching variable name from 'scondust' (surface dust concentration) to 'mmrdust' (dust mass ratio at all model layers), even though only one of them (NorESM2-LM) has ensembles (two members). We added these six models to the evaluation with the surface dust extracted from the lowest model layer. All the related texts and figures were updated and GISS-E2-1-G still performs the best in capturing the drought-dust sensitivity when the SEUS is under droughts.

**Reviewer #2**
General Comments:
Fine surface dust in the southeastern U.S. is known to increase during summer months due to long-range transport of North African dust to the region. This manuscript investigates dust-drought relationships in the region and changes in large-scale atmospheric variability and teleconnections with drought and dust transport to the SEUS. Evaluation of global transport models against observations also elucidates the ability of models to capture these connections during severe drought periods. The manuscript is well-organized and written, and is an important contribution to the literature. I suggest publishing after minor corrections based on the comments below.

Specific Comments:
Line 97: Can the authors provide more details regarding completeness criteria for including data from these sites? How do the different sampling periods at some sites (6-day vs 3-day) affect daily interpolations? Also, additional sites come online during this period (2000-2019), did the authors only use sites that were operating during the entire period? Adding sites for different years could bias the results from year to year. How did the authors treat the bias between the CSN and IMPROVE dust concentrations when combining the data? (e.g., Hand et al., 2012; Gorham et al., 2021).
**Response**: Thanks for pointing out the dust measurement bias between the CSN and IMPROVE network. Indeed, Hand et al. (2012) reported a 32% lower dust concentration at the collocated IMPROVE sites than CSN sites and concluded the urban-to-rural dust comparison should be approached with caution. We have removed the CSN sites from the analysis and the revised paper uses IMPROVE sites only. Since IMPROVE sites are sampled every 3 days consistently across all sites, the sampling frequency is not an issue anymore.

To investigate the effect of changes in sites as the reviewer pointed out, we compared three ways of treating IMPROVE data: (1) using all the available IMPROVE sites (IMPROVE_raw), (2) using IMPROVE sites with at least a 5-year (IMPROVE_5year) data record, and (3) using

IMPROVE sites with at least a 10-year data record (IMPROVE_10year). We examined the mean differences and interannual correlation coefficients (R) between the three datasets in Figure R4. Higher mean differences and lower R values can be seen between the IMPROVE_5year and IMPROVE_raw datasets over part of Georgia, Nevada, and Washington states, where the IMPROVE sites with less than 5-year data are located. By contrast, the IMPROVE_5year and IMPROVE_10year datasets have a good agreement, as suggested by the near-zero mean differences and near-one R over almost all the grids. Thus, we chose to use the IMPROVE_5year dataset and redid all the associated analyses. We added some texts in Line 98-110 to explain this and inserted Figure R4 in the supplement file as Figure S1.

(a) Mean difference of dust during 2000-2019 JJA

[Figure]

(b) Interannual correlation coefficient (R) of dust during 2000-2019 JJA

[Figure]

**Figure R4**: Dust mean differences (a) and interannual correlation coefficients (b) between the datasets interpolated from the IMPROVE sites with a data record of more than 5 years (IMPROVE_5year) and all the IMPROVE sites with data available (IMPROVE_raw) during the study period (left column), and between the IMPROVE sites with a data record of more than 10 years (IMPROVE_10year) and IMPROVE_5year (right column). N_IMPROVE in a indicates the number of sites (black circles) used for the IMPROVE_5year (left) and IMPROVE_10year (right) datasets, respectively. Red circles in a show the sites used for the IMPROVE_raw dataset but not for the IMPROVE_5year (left; 23 sites), and for the IMPROVE_5year dataset but not for the IMPROVE_10year (right; 13 sites). These red circles are included to help better understand the changes.

Line 117: From 1996 until when?
**Response**: From 1996 until the present. The sentence was modified accordingly. We used the period of 2000-2019 JJA as denoted in Table S1.

Line 134: If am I understanding correctly, under extreme drought conditions the data for each site could correspond to different days?
**Response**: Yes. The drought conditions of the sites and grids are spatially and temporally matched with USMD data.

Line 157: The shifts in Figure S1 appear different for both CONUS (severe drought is shifted further) and for the SEUS, it is not shifted as much. Can the authors elaborate?

**Response**: For the CONUS, the dust changes at the IMPROVE sites are similar between Figure 1 (USDM-based) and Figure S1 (SPEI-based) with an increase of the mode and mean value by ~0.13 µg/m$^3$ and ~0.3 µg/m$^3$, respectively. It appears that the monthly data shifts further because its narrower x-axis range makes the change more evident. However, the dust changes are indeed smaller in the SEUS on a monthly scale than on a weekly scale, which can be explained by the episodical nature of the African dust transport events. It typically takes about ten days for the African dust to reach the SEUS (Chen et al., 2018; Pu and Jin, 2021). Thus, weekly data is supposed to capture such events better than monthly data, in which high dust values will be averaged out by low values when no dust event occurs. To further explain this, we examine the weekly and monthly time series of dust at the OKEF1 site in the SEUS (Figure R5). Dust on a weekly scale shows higher variabilities than that on a monthly scale with the maximum dust reaching 10 µg/m$^3$. When a fast-developing drought changes severity within a month, the monthly dust average will dampen the difference between drought and non-drought conditions. We added this discussion to Line 161-164.

[Figure]

**Figure R5**: Weekly (top) and monthly (bottom) time series of dust at the OKEF1 (82.12ºW,30.74ºN) IMPROVE sites. Dots in the weekly time series are color-coded by USDM.

Line 170: I am not sure I follow the reasoning for conducting the linear regression only using the lowest four SPEI bins. It would seem that the reasoning for doing this should apply to both the west and the east. Otherwise, it appears the data points are being ignored to get the desired results.

**Response**: As suggested by the other reviewer (2[nd] comment), we recalculated the slopes using drought conditions only (SPEI < 0) at each grid box in Figure R1c. More SPEI bins were added as well to increase the data points used for regression. Most of the grids in the western US still show negative slopes and more grids in the SEUS show positive slopes compared to the results of using all the data. More texts were added in Line184-199 of the revised manuscript to better explain this.

Line 187: How did the authors determine how the southeast region is defined with the box shown in Figure 1a? How did they decide on the lat/lon limits or sites to include?
**Response**: The selection of the southeastern region boundary is to cover most of the grids/sites with negative dust changes under severe droughts as shown in Figure 1a. On the other hand, the boundary cannot be extended to the further north because African dust barely reaches the northern US where local or even Asian dust plays a more important role (Aldhaif et al., 2020). To better explain this, we compared the dust changes under non-drought and severe drought after expanding the northern boundary to 36ºN and 39ºN, respectively (Figure R6). Dust shows little changes in the boundary of [100ºW-75ºW; 25ºW-36ºW] and increases in the boundary of [100ºW-75ºW; 25ºW-39ºW]. This further verifies that our boundary selection is correct to capture the reduced dust signal. The explanation was added to Line 154-157 in the manuscript with Figure R6 inserted as Figure S2 in the supplement file.

[Figure]

**Figure R6**: Boxplots of dust using three different delimitations of southeastern US (SEUS) under wet and normal (non-drought) and severe drought conditions.

Line 207: How were these particular limits chosen?
**Response**: The same question was also asked by the other reviewer, and we reproduce the responses below.

We examined the percentage of grids covered by D2-D4 over the SEUS in increasing order (Figure R7a). There appears to be a 'turning point' around 30%, after which the percentage increases much faster. This indicates a regional expansion of severe drought events. Therefore, we selected 30% as the threshold of regional severe drought events. Most of the weeks have 30%-60% of grids under N0 or D0-D1. In some weeks, the percentage of grids under N0 and D0-D1 can be quite close (e.g., 50% and 47% ). To avoid the inclusion of these weeks into the non-drought conditions and reduce the impact of mild drought (D0-D1), we decided to set 70% as the threshold of regional non-drought events. The threshold of 30% of the grids under D2-D4 is nearly at the top 20% quantile of all the weeks. To match this, we select the 20% quantile of regional-mean SPEI as the threshold for regional severe drought events on a monthly scale. Months with regional-mean SPEI greater than the 20% quantile are considered as non-drought events.

We also tested the results of using other thresholds by reproducing the differences of MODIS AOD (Figure R7b; USDM-based) and dust extinction coefficient (Figure R7c; SPEI-based)

between severe drought and non-drought events the same as those in Figure 4 (new Figure 5). The signal of reduced dust under droughts is consistent, which means the threshold selection does not significantly affect our conclusions. These explanations were added in the main text between Line 233-245. Figure R7 was also included in the supplement file as Figure S4.

[Figure]

**Figure R7**: (a) Percentage of grids under severe drought (D2-D4; red dots) in increasing order and the corresponding percentage of grids under non-drought (N0; blue dots) and mild drought (D0-D1; orange dots). (b) MODIS AOD difference between severe drought and non-drought events if more than 40% grids are under D2-D4 (severe drought) and more than 60% grids are under N0 (non-drought; left column), or more than 20% grids are under D2-D4 (severe drought) and more than 80% grids are under N0 (non-drought; right column). (c) Dust extinction coefficient difference between severe drought and non-drought events if the regional-mean SPEI is greater (severe drought) and smaller (non-drought) than 10% quantile (left column) or 30% quantile (right column). Black or orange dots in b and c indicate the significant difference at a 95% confidence level.

Line 207: It also helps clarity of writing to include the opposite description in the text and not just parentheses (here and other places in the manuscript and captions), such as "Regional severe drought (non-drought)". Unless there are page limits and space issues, it causes more effort to understand than to just write it out.

**Response**: Good suggestion and we revised the related sentences and figure captions.

Line 212: Are these droughts limited to the SEUS region mentioned above?

**Response**: Yes. We specified this in the sentence.

Line 214: How was this AOD limit chosen?
**Response**: The AOD value of 0.15 is about 20% quantile of all the values in the red box of Figure 4b (new Figure 5a). After a second thought, we decided not to specify a value and changed the sentence to 'higher AOD values than its surroundings' in Line 258.

Line 259: Can the authors comment regarding the years with severe droughts that aren't associated? Such as 2000, 2006, and 2019?
**Response**: The reason for the years that are not closely associated with strong negative NAO is that NAO is not the only factor causing drought conditions in the SEUS. Another factor widely reported by previous studies is the cold phase of El Niño and the Southern Oscillation (ENSO), also known as La Niña (e.g., Piechota and Dracup, 1996; Cook et al., 2007). La Niña in 2000 and 2006 is linked with the fast-developing droughts in the SEUS by Chen et al. (2019) despite the NAO index is not too strong. In summer 2019, ENSO was in its warm phase (CPC, 2022), which may counteract the effects of negative NAO and result in non-drought conditions. By contrast, both La Niña and negative NAO occurred in 2011 and cause severe drought events in the SEUS (Pu et al., 2016). Although many other factors contribute to the SEUS droughts, the abnormal circulation patterns related to the negative phase of NAO are closely linked with the African dust transport, and thus we focus on NAO in this study. These explanations were added to Line 323-330 in the revised manuscript.

Line 319: Include units with 4.76.
**Response**: Done. We also rewrote the model evaluation section by adding more models suggested by the other reviewer (5th comment).

Line 331: Typo: "unite"
**Response**: Done.

Line 355: It may be less confusing to write this as the conditions that caused reduced transport of dust also correspond to periods with drought conditions (so it doesn't seem that drought conditions in the SE are somehow causing less dust transport from Africa). The description in line 359-361 clarifies this but it could be misinterpreted here.
**Response**: We changed the sentence to "we investigated how drought conditions in the SEUS can be linked with the trans-Atlantic transport of African dust" so that it does not seem to imply a causal relationship.

Figure 1: Include JJA in the caption for part (a). Are the results shown in part (b) also for JJA?
**Response**: Yes. Both are during JJA months, and we added the JJA period to the caption.

Figure 2: Typo for part (b): "reginal". Do these results correspond to all time periods?
**Response**: The typo was corrected. The time period is 2000-2018 JJA, which is the shared period between dust and SPEI data. We added this time period in the caption.

Figure 3: Why were only IMPROVE data used (not CSN)?
**Response**: We changed to use IMPROVE sites only in the revised manuscript.

Figure 4: Typo in legend of part (a): "sever". Also, typo in caption: "Carrabin". Include the wavelength of AOD in the caption.
**Response**: The typos are corrected and the wavelength of AOD (550nm) was also added.

Figure 5: Are these changes based on severe droughts only in the SEUS region? What years are included in this figure?
**Response**: Yes. These changes are based on the USDM-based regional drought events during 2000-2019 JJA as displayed in the time series of new Figure 4. The period was added in the caption.

Figure 6: Include time periods in caption.
**Response**: Done.

Figure 7: Include time periods in caption.
**Response**: Done.

Figure 8: Line 306 states JJA but this legend reads "all months". What is the time period? Include time period in caption. Also include that the dashed line corresponds to 1:1.
**Response**: "JJA months" is the correct one. The legend was changed to 'all JJA months'. The time period is 1973-2014 JJA, which was added to the caption together with the 1:1 dashed line.

Table 1: Include time period in caption
**Response**: Done.

[revised manuscript text omitted]